# A simple generative model of the mouse mesoscale connectome

**Sid Henriksen[1,2*†], Rich Pang[3*†], Mark Wronkiewicz[3*†]**

[1]Laboratory of Sensorimotor Research, National Eye Institute, National Institutes of Health, Bethesda, United States; [2]Institute of Neuroscience, Newcastle University, Newcastle upon Tyne, United Kingdom; [3]Graduate Program in Neuroscience, University of Washington, Seattle, United States

**Abstract** Recent technological advances now allow for the collection of vast data sets detailing the intricate neural connectivity patterns of various organisms. Oh et al. (2014) recently published the most complete description of the mouse mesoscale connectome acquired to date. Here we give an in-depth characterization of this connectome and propose a generative network model which utilizes two elemental organizational principles: proximal attachment – outgoing connections are more likely to attach to nearby nodes than to distant ones, and source growth – nodes with many outgoing connections are likely to form new outgoing connections. We show that this model captures essential principles governing network organization at the mesoscale level in the mouse brain and is consistent with biologically plausible developmental processes.

**\*For correspondence:** sid. henriksen@gmail.com (SH); rpang@uw.edu (RP); wronk@uw. edu (MW)

†These authors contributed equally to this work

**Competing interests:** The authors declare that no competing interests exist.

## Introduction

The network of physical connections among neurons in the brain provides the medium for neural communication. Investigations of these anatomical networks are typically categorized as macro-, meso-, or microscale, depending on the spatial resolution of the techniques used. The mesoscale, which describes the connectivity among local populations (hundreds to thousands) of neurons, is an attractive intermediate between the two more extreme scales: it has higher granularity than macroscale data, which details connectivity between large anatomically defined brain areas, but it has a broader lens than the microscale, which is concerned with synaptic level connections, often in relatively small volumes of tissue (see (*Sporns et al., 2005*) for review). These properties make the mesoscale tractable enough for whole-brain (i.e. connectomic) studies with current technological and analytical tools.

The Allen Institute for Brain Science recently constructed a mesoscale connectome for the mouse (the Allen Mouse Brain Connectivity Atlas), which was the first complete connectivity dataset of a mammalian brain at the mesoscale (*Oh et al., 2014*). Using injections of an anterograde fluorescent viral tracer and serial two-photon microscopy, *Oh et al. (2014)* comprehensively mapped both intra- and interhemispheric axonal tracts and estimated the directed connectivity structure among 213 non-overlapping anatomical regions. The authors also conducted a preliminary graph theoretic analysis and showed that basic network properties of the mouse connectome could not be explained by any one standard network model (*Oh et al., 2014*). *Rubinov et al. (2015)* extended this analysis by identifying small-worldness, a hierarchical modular structure, and non-optimal wiring in the connectome (*Rubinov et al., 2015*). As the mouse is one of the most pervasive model organisms in biomedical science, a deeper characterization of its connectome is likely to provide pertinent groundwork for future studies of brain development and function as well as yield insights into the broader organizational principles of the mammalian brain.

**eLife digest** Within the brain, neurons organize themselves into extensive networks. The physical connections between neurons determine which groups of neurons are able to communicate with one another. Recently, researchers mapped out the neural circuits within the entire brain of an adult mouse. The resulting wiring diagram, or 'connectome', provides an opportunity to study these brain networks in unprecedented detail.

One key question is how these networks acquire their structure. Henriksen, Pang and Wronkiewicz wondered whether the patterns of connections in the adult mouse brain might provide clues to how this connectivity emerges. Analyzing the adult mouse connectome revealed a number of unexpected properties. For example, a brain region's incoming connections were often different from its outgoing connections. This suggests that it is important to consider the direction of connections between groups of neurons, and that different mechanisms may govern how incoming and outgoing connections are formed. Furthermore, if a brain region was connected to only a few others, those regions tended to also be connected among themselves.

Using this information, Henriksen, Pang and Wronkiewicz attempted to 'grow' a virtual mouse connectome from scratch using two simple rules deduced from the properties of the adult connectome. The first rule was that brain regions with many outgoing connections are more likely to form more outgoing connections. The second was that outgoing connections are more likely to connect to nearby brain regions than to distant ones.

The resulting model successfully reproduced a number of key properties of the mouse brain connectome. This suggests that relatively simple principles help to determine at least some of the structure of networks within the adult brain. The next challenge is to identify the exact relationships between these principles and the biological mechanisms that support brain development.

Graph theory provides a mathematical framework for investigating the organization of networks and has been increasingly applied in neuroscience over the last 15 years. Graphs are mathematical objects that consist of nodes and connections between the nodes, called edges (*Bullmore and Sporns, 2009*; *Rubinov and Sporns, 2010*). Edges can be either directed or undirected, as well as binary or weighted. In the Allen mouse connectome, nodes and edges correspond to brain regions and axonal tracts, respectively. Conventionally, real-world networks (such as the World Wide Web or social networks) are compared to binary undirected graphs such as small-world (*Watts and Strogatz, 1998*) and scale-free graphs (*Barabasi and Albert, 1999*). These models have been critical for understanding the conditions under which various network properties, such as small-worldness (high clustering among nodes combined with short average path lengths between node pairs) and scale-freeness (defined by power-law degree distributions), emerge in the brain and other systems.

Modeling brain networks with these standard graphs, however, requires some limiting simplifications. First, because they are not embedded in physical space, these networks ignore the biological cost of constructing physical fiber tracts, as well as spatial constraints imposed by the surrounding tissue. Second, small-world and scale-free models ignore directionality, intrinsically discarding information about differences in incoming and outgoing connection patterns. Both properties are crucial considerations when modeling the brain (*Laughlin and Sejnowski, 2003*; *Song et al., 2014*; *Kaiser et al., 2009*; *Ercsey-Ravasz et al., 2013*). Third, these models were either not developed with neural data (*Barabasi and Albert, 1999*) or used data from simple model organisms, such as *Caenorhabditis elegans*, at a particular spatial scale (*Watts and Strogatz, 1998*), which constrains their relevance to understanding connectomes of other organisms. For these reasons, the field has recently turned to exploring non-standard network models to elucidate generative principles of real brain networks.

Recent work in developing plausible generative network models for the brain has primarily addressed spatial embedding (*Song et al., 2014*; *Kaiser et al., 2009*; *Ercsey-Ravasz et al., 2013*; *Klimm et al., 2014*; *Kaiser and Hilgetag, 2004*). While the exact approaches differ in their implementation and scale of the networks being modeled, a common theme is that a node is more likely to connect to nearby nodes than distal ones. This organizational principle has been able to capture a

range of properties observed in the cortex, including the distribution of connection lengths (*Song et al., 2014*; *Kaiser et al., 2009*; *Ercsey-Ravasz et al., 2013*), the inverse relationship between degree and clustering coefficient (*Watts and Strogatz, 1998*; *Song et al., 2014*; *Betzel et al., 2015*; *Mitra, 2014*), and the relative frequency of three-node motifs (*Ercsey-Ravasz et al., 2013*). Additional generative rules have been explored by *Klimm et al. (2014)*, and while the resulting models have captured many properties of cortical networks, the authors note that these rules likely do not reflect the underlying generative principles of cortical networks. Similarly, *Betzel et al., (2015)*, recently reported that individual human macroscale connectomes are well-fitted by generative network models, which use both spatial proximity and homophilic attraction (i.e. nodes with similar graph theoretic properties are more likely to form connections). However, the homophilic rules employed by Betzel et al. also do not lend themselves to straightforward biophysical interpretations. Indeed, the difficulty of developing biophysically interpretable rules is a recurring challenge in generative network models (*Watts and Strogatz, 1998*; *Song et al., 2014*; *Klimm et al., 2014*; *Betzel et al., 2015*; *Vértes et al., 2012*).

Here, we provide an in-depth analysis of the mouse connectome's properties and use the findings to develop a generative network model of the mesoscale connectome. We characterized the directed and undirected degree distributions, clustering coefficient distribution, reciprocity, global efficiency, physical edge length distribution, nodal efficiency, and the characteristic path length of the connectome (Table 2 in 'Materials and methods' for definitions or *Bullmore and Sporns, 2009* and *Rubinov and Sporns, 2010* for review). Informed by these data, we developed a spatially embedded directed network model. This model uses two simple generative principles: proximal attachment (PA) – outgoing connections are more likely to attach to nearby nodes than distal ones, and source growth (SG) – nodes with many outgoing connections are more likely to develop new outgoing connections. We show that this simple model, parameterized only by a length constant and the number of nodes and edges, can capture directed, undirected, and spatial properties of the mouse connectome. This work supports the existing literature on the importance of spatial embedding and provides strong evidence that SG is a major phenomenological rule that shapes connectivity patterns in the mouse brain. Lastly, we propose biological mechanisms that might account for these two generative principles.

## Results

We analyzed the Allen Mouse Connectivity Atlas (*Oh et al., 2014*), which is the most comprehensive mesoscale connectome collected to date. We used the linear model from *Oh et al. (2014)* to build an adjacency matrix containing connections between 213 symmetric pairs of nodes (426 total) and 8820 directed edges (7804 undirected).

### Comparison of the connectome to undirected graph models

We first compared the undirected structure of the mouse connectome with that of three well-characterized standard graphs commonly used in the literature: a degree-controlled random network (*Maslov and Sneppen, 2002*), a small world network (*Watts and Strogatz, 1998*), and a scale-free network (*Barabasi and Albert, 1999*). The mouse connectome is characterized by a degree distribution with many low-degree nodes and a long tail of high-degree nodes (*Figure 1a*; (*Oh et al., 2014*)). The degree distribution was not well replicated by any standard graph (*Figure 1a and b*), nor was the clustering coefficient distribution – a finding also shown in *Oh et al. (2014)*. Although the scale-free network's degree distribution most closely resembles that of the connectome, by construction, it cannot capture the distribution of low-degree nodes (as all nodes are instantiated with a minimum number of edges).

The structural discrepancy between the mouse connectome and standard networks was further exposed by examining the relationship between degree and clustering coefficient (*Figure 1c–f*). In the connectome, nodes with lower degree tend to be more clustered. This relationship was well-fitted with a power law: $C_i \propto k_i^{\gamma}$, where $C_i$ and $k_i$ denote clustering coefficient and degree, respectively, and with the best-fit $\gamma = -0.44$. Previous studies have shown that such a relationship (with varying $\gamma < 0$) is common to many real-world networks (*Ravasz and Barabási, 2003*), including the human connectome (*Klimm et al., 2014*). While the small-world network exhibits a clustering coefficient distribution similar to that of the connectome (*Oh et al., 2014*), as well as an inverse

relationship between clustering coefficient and degree, these similarities are superficial: inspecting *Figure 1e* reveals that the small-world network's degree distribution is much more homogeneous. While the scale-free network partly captures the connectome's degree distribution, its nodes have a much lower clustering coefficient, and there is a weaker relation between these two metrics (*Figure 1f*). The degree-controlled random model shows that shuffling the connectome's edges yields a graph resembling the scale-free network (*Figure 1d*). Thus, these graphs fail to capture key aspects of the connectome's undirected structure.

Previous modeling efforts have shown that several properties of brain networks can be captured by simple models, which assume that spatially nearby nodes are more likely to connect than distal ones (*Song et al., 2014*; *Kaiser et al., 2009*; *Ercsey-Ravasz et al., 2013*; *Kaiser and Hilgetag, 2004*). Therefore, we explored an undirected spatially embedded network model. We first assigned to each node a spatial position randomly sampled from a 7 mm x 7 mm x 7 mm cube, which gave an inter-nodal distance distribution similar to the connectome (*Figure 5—figure supplement 1*). Edges were then added between pairs of nodes $i$ and $j$ by choosing node $i$ at random and node $j$ with probability $P_{ij} \propto exp(-d_{ij}/L)$. That is, the probability of choosing a target node $j$ decayed with the Euclidean distance $d_{ij}$ between nodes $i$ and $j$ according to length constant $L$ (see 'Materials and methods'). We call this rule proximal attachment (PA).

Relative to the connectome, the purely geometric model exhibited relatively narrow Gaussian-like degree distributions across several values of $L$ (*Figure 2a*). Thus, the PA rule fails to generate low- and high-degree nodes. This geometric model does, however, exhibit an inverse relationship between degree and clustering (*Figure 2b*) as in the connectome. However, *Figure 2b* demonstrates that there is no value of $L$ for which this model adequately captures the joint degree-

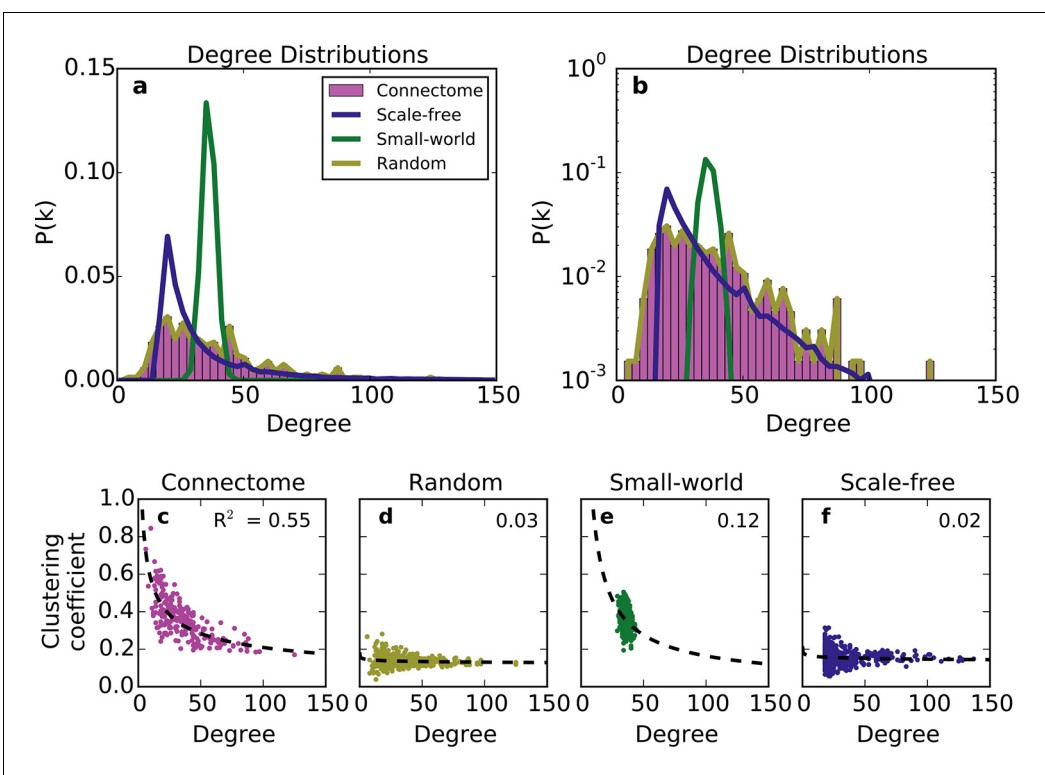

**Figure 1.** Standard graph models vary in their ability to recreate the mouse connectome's degree distribution and relationship between degree and clustering coefficient. (**a-b**) Probability density of degree distributions for the mouse connectome, and an average over 100 repeats of scale-free, small-world, and (degree-controlled) random networks plotted with (**a**) linear and (**b**) log-linear scales. (**c-f**) Clustering coefficient as a function of degree for each node in (**c**) the mouse connectome, (**d**) a degree-controlled random network, (**e**) a small-world network, and (**f**) a scale-free network. Each plot shows data from 426 nodes and the best fitting power law function (dashed line). (**a**) is similar to Figure 3c from *Oh et al. (2014)*.

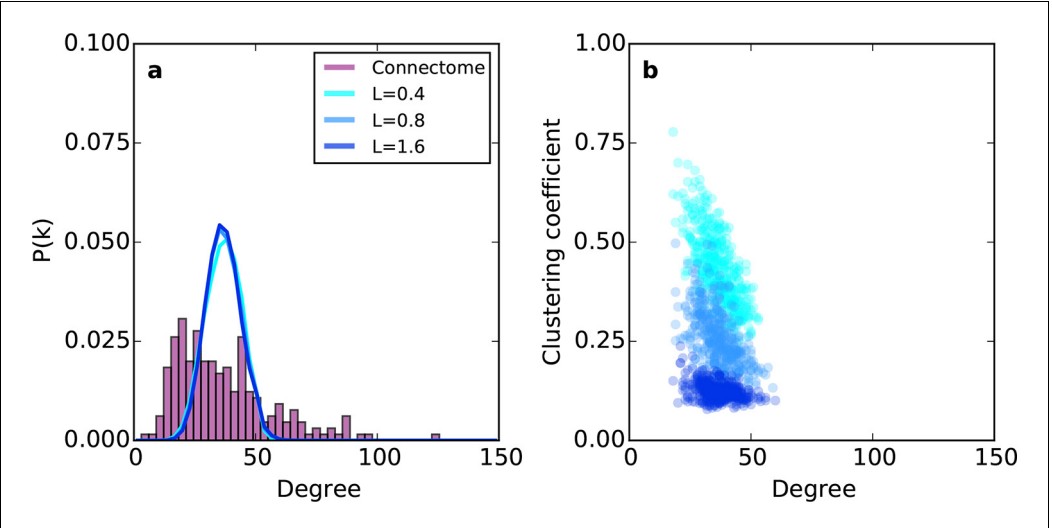

**Figure 2.** Example networks generated using the (purely geometric) proximal attachment (PA) rule where connection probability between two nodes decreased with distance. (a) The degree distribution for networks grown with three values of L (in mm), each averaged across 100 repeats. The mouse connectome is shown for comparison. (b) Clustering coefficient as a function of degree for representative networks grown with the same values of L used in (a).

clustering distribution seen in the connectome; the PA model misses both low-degree nodes with high clustering and high-degree nodes with low clustering. This suggests that the purely geometric rules explored in previous studies (*Song et al., 2014*; *Kaiser et al., 2009*; *Ercsey-Ravasz et al., 2013*; *Kaiser and Hilgetag, 2004*) are not sufficient to recreate the mouse connectome's properties. Therefore, we explored the hypothesis that topological rules, which are based on properties of individual nodes, play an important role in forming the connectivity patterns of the mouse connectome.

## The directed perspective and novel random graphs

Conventionally, topological rules in generative network models have been applied to undirected networks. However, the mouse connectome contains directed edges, allowing us to probe whether the directionality of connections in the mouse connectome plays a role in shaping connectivity patterns. In a directed graph, edges point from source nodes to target nodes, so one can consider each node's in-degree and out-degree (i.e. the total number of incoming and outgoing connections for a node, respectively). We found a surprisingly asymmetric relationship between these distributions in the connectome – while the in-degree of the network was approximately normally distributed, the out-degree distribution exhibited a peak near zero and a long tail (*Figure 3a*), much like an exponential distribution. There was no significant correlation between in- and out-degree ($\rho = 0.127$, p = 0.065, Spearman rank correlation). *Figure 3b* shows the proportion of incoming edges as a function of total degree for all nodes in the mouse connectome. As the total degree of a node increases, the proportion of incoming edges decreases.

To model the directed mouse connectome, we extended the purely geometric PA model by considering two mathematically symmetrical, yet phenomenologically distinct frameworks: target attraction and source growth (TAPA and SGPA). When adding an edge in the TAPA model, a *target* node is chosen with a probability proportional to its in-degree, while the *source* node is chosen with a probability that decreases with its distance from the target node (*Figure 4*, top). In the SGPA model, a *source* node is chosen with a probability proportional to its out-degree, while the *target* node is chosen with a probability that decreases with its distance from the source node (*Figure 4*, bottom). The TAPA and SGPA models can be considered directed spatial variants of the preferential attachment algorithm introduced by *Barabasi and Albert (1999)* and lead to "rich-get-richer" patterns of either incoming or outgoing connection formation. In both cases, all nodes were initialized upon network instantiation (unlike in the preferential attachment algorithm where nodes are added

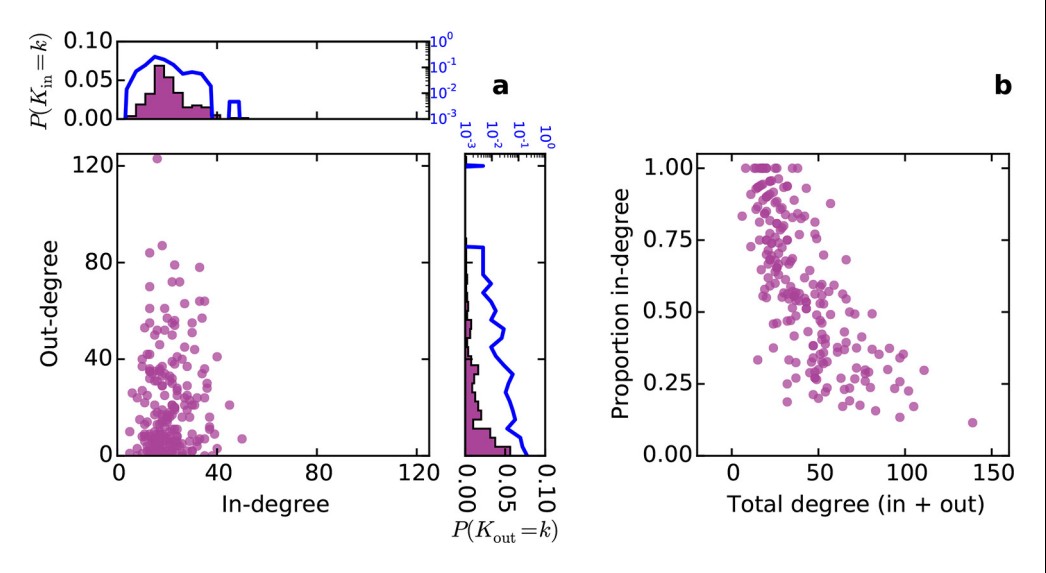

**Figure 3.** Directed analysis of the mouse connectome reveals different distributions for in- and out-degree. (a) Out-degree as a function of in-degree for all nodes in the mouse connectome. Margins show in- and out-degree distributions with the blue lines and axis labels corresponding to a logarithmic scale. In-degree is approximately normally distributed while the out-degree approximately follows an exponential distribution. (b) Proportion in-degree as a function of total degree (i.e. in-degree divided by the sum of incoming and outgoing edges). Low-degree nodes tend to have mostly incoming edges, whereas high-degree nodes are characterized by mostly outgoing edges.

The following figure supplement is available for figure 3:

**Figure supplement 1.** Distribution of proportion in-degree in the mouse connectome.

---

iteratively), and directed edges were added iteratively until the number of edges matched the mouse connectome. Since these models also incorporated the geometric PA property, both models were parameterized by a single free parameter $L$ exactly as in the PA model.

## Comparison of the connectome to the directed graph models

The SGPA model exhibits an in-degree distribution that is approximately normal and an out-degree distribution that is approximately exponential (*Figure 5a*, cyan). This is matches the mouse connectome (*Figure 3a*) but is exactly opposite for the TAPA model (*Figure 5a*, red). *Figure 5b* shows proportion in-degree as a function of total degree (in-degree + out-degree) for both network models. In the SGPA model (and the mouse connectome; *Figure 3b*), high-degree nodes tend to have a large proportion of outgoing connections. Again, this is opposite of the TAPA model, where high-degree nodes tend to have a large proportion of incoming connections. These results suggest that the directed graph theoretic properties of the mouse connectome are best captured by a SG model of network generation.

Previous work has shown that both the reciprocity coefficient and average clustering coefficient of the brain are well above chance (*Oh et al., 2014*; *Rubinov and Sporns, 2010*; *Felleman and Van Essen, 1991*; *Kaiser and Varier, 2011*). We found that the mouse connectome has a reciprocity coefficient of 0.13, which is high compared to that expected from chance (0.03). Reciprocal edges in the connectome are on average shorter than nonreciprocal edges (*Figure 5c*; $t(8818) = 13.25$, p << $10^{-10}$, independent samples t-test; Cohen's d = 0.47). As shown in *Figure 5e*, the reciprocity coefficient of our model networks decreases with increasing length constant, approximately matching that of the mouse connectome (0.13) when $L = 0.725$ mm for both the SGPA and TAPA models. However, fitting the length constant to reciprocity (e.g. in the SGPA model) underestimates the physical edge lengths of the connectome (compare *Figure 5c to d*). Using a larger length constant improved

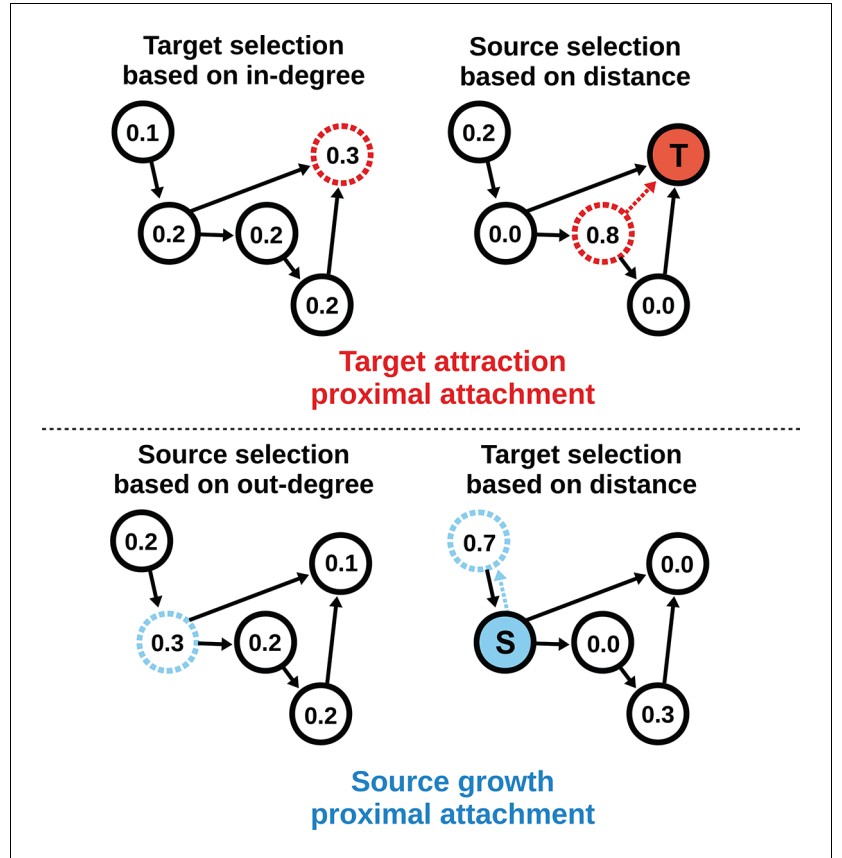

**Figure 4.** Target attraction and source growth network generation algorithms. The numbers inside each node indicate the probability of growth or attachment. For illustration, the most probable node (dashed) is selected in both diagrams (T and S, corresponding to target and source, respectively). Top row: target attraction proximal attachment (TAPA) model. A target node is selected with a probability proportional to its in-degree (left), while the source node is chosen with a probability that decreases exponentially with the node's Euclidean distance from the target (right). Two nodes have zero probability of forming an edge since the target is already receiving projections from these nodes. The dashed red line shows the most probable edge. Bottom row: source growth proximal attachment (SGPA) model. A source node is selected with a probability proportional to its out-degree (left), while the target node is chosen with a probability that decreases exponentially with the node's Euclidean distance from the source (right). The dashed cyan line shows the most probable edge. In both algorithms, we assume that each node begins with a self-connection (corresponding to an outgoing and incoming edge) to avoid zero-valued probabilities, though self-connections are not shown here or included when calculating any metrics.

the fit to the reciprocal and nonreciprocal edge length distributions (not shown), but reduced the model's reciprocity.

Incorporating SG into the PA model increases the width of the degree distribution, allowing the model to also capture the joint clustering-degree distribution seen in the connectome (*Figure 6a*). A network grown solely with the PA rule and a random degree-controlled network both showed a joint clustering-degree distribution that differed from the connectome's, again suggesting that both topological and geometric rules are important when modeling the connectome. The inset in *Figure 6a* quantifies this by showing that the SGPA model's fitted power law relationship between clustering coefficient and degree is the most similar to the connectome. Additionally, this increased clustering is spatially localized: In the connectome, there is a negative correlation between a node's clustering coefficient and its mean edge length. That is, nodes with short average edge lengths tend to be highly clustered ($\rho$ = -0.238, $p < 10^{-6}$, Spearman rank correlation). A similar, but stronger relationship occurs in the SGPA model (median $\rho$ = -0.482, $p < 10^{-10}$, Spearman rank correlation).

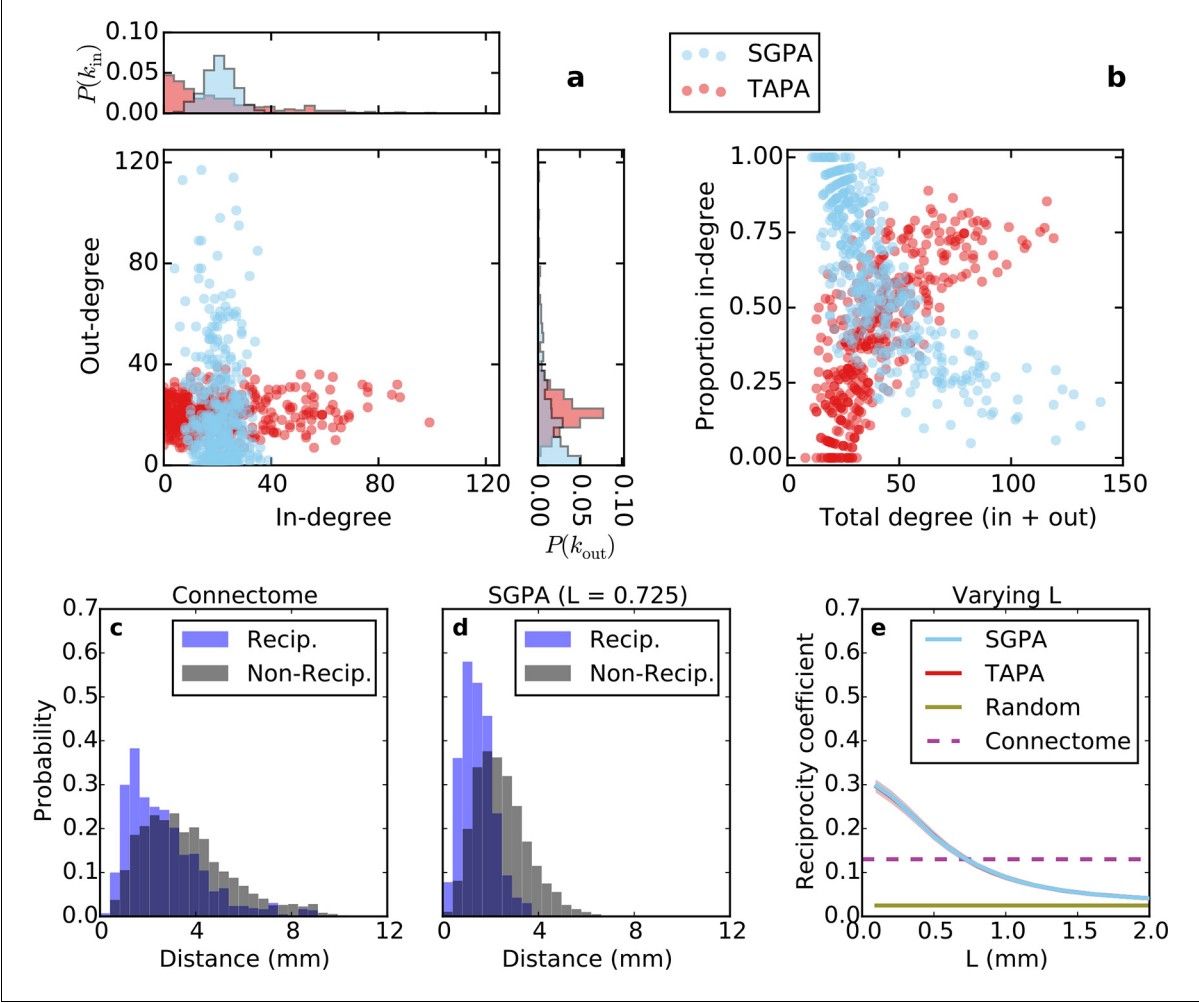

**Figure 5.** Directed analysis of single representative TAPA and SGPA network models and reciprocity comparison with the connectome. (a) Out-degree as a function of in-degree for both algorithms with $L = 0.725$ mm, which was chosen to match the connectome's reciprocity – see e). Margins show in- and out-degree distributions. (b) Proportion in-degree as a function of total degree for both algorithms. The SGPA model qualitatively captures the connectome's directed degree distributions and proportion in-degree (cf. *Figure 2*). (c) Edge length distribution for the connectome, shown for both reciprocal (blue) and non-reciprocal edges (black). (d) Same as (c) but for the SGPA model. (e) Reciprocity coefficient as a function of the length parameter for both TAPA (red) and SGPA (cyan), with shading indicating standard deviation over 100 repeats. Both models intersect the connectome at $L = 0.725$ mm. For reference, the reciprocity coefficient for the connectome (magenta) and a corresponding degree-controlled random graph (gold) are also shown. SGPA and TAPA models overlap.

The following figure supplements are available for figure 5:

**Figure supplement 1.** Directed degree distributions and proportion in-degree for a directed Erdos-Renyi graph.

**Figure supplement 2.** Directed degree distributions and proportion in-degree for a purely topological source-growth or target-attraction directed graph (SG only, TA only, respectively), with no proximal attachment (equivalent to $L = \infty$ in SGPA or TAPA).

**Figure supplement 3.** Directed degree distributions and proportion in-degree for an SGPA model where the network is grown one node at a time.

**Figure supplement 4.** Inter-nodal distance distribution for the mouse brain (magenta bars) and a 7 mm³ cube (black line).

**Figure supplement 5.** Directed degree distributions and proportion in-degree for a graph in which source selection was proportional to total degree (in-degree + out-degree) in (a) and (b), and in which source selection was proportional to total degree raised to the power $\gamma = 1.67$ in (c) and (d).

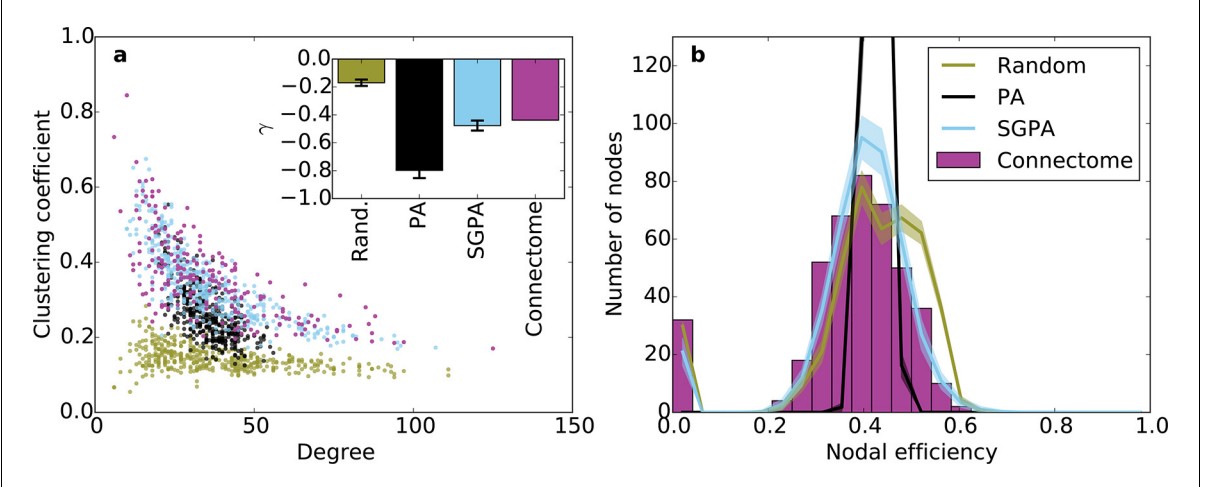

**Figure 6.** Clustering and nodal efficiency for the connectome and other directed models. (a) Clustering-degree joint distribution for connectome and one representative instantiation of each model graph. Inset: best-fit power-law exponent $\gamma$ (see 'Materials and methods') for the connectome ($\gamma$ = -0.44, $R^2 = 0.58$) and random (mean $\gamma \pm$ std. = -0.17 $\pm$ 0.02, median $R^2 = 0.16$), SGPA (mean $\gamma \pm$ std. = -0.48 $\pm$ 0.04, median $R^2 = 0.59$), and geometric PA (mean $\gamma \pm$ std. = -0.80 $\pm$ 0.06, median $R^2 = 0.42$) models. (b) Distributions of nodal efficiencies (see 'Materials and methods') with mean $\pm$ standard deviation (line and shaded regions, respectively) for model networks. Mean $\pm$ standard deviation of mean nodal efficiency (averaged over nodes) is 0.409 $\pm$ 0.001 for the random model, 0.393 $\pm$ 0.004 for the SGPA model, and 0.426 $\pm$. 001 for the pure geometric model. The connectome's average nodal efficiency is 0.375. The PA model's histogram peaks at 279 at a nodal efficiency of 0.44. (b) and the inset in a) both used 100 sample instantiations of each model.

The following figure supplements are available for figure 6:

**Figure supplement 1.** Clustering and nodal efficiency for Erdos-Renyi (ER) and TAPA models.

**Figure supplement 2.** Undirected degree distribution for the SGPA model and the mouse connectome in (a) linear and (b) logarithmic scales.

**Figure supplement 3.** Clustering vs. degree (a) and nodal efficiency (b) for the node-by-node SGPA network (L = 0.725) used in *Figure 5—figure supplement 3*.

## Higher order connectivity statistics

Our previous analyses of degree, in-degree, out-degree, and clustering coefficient distributions describe the connectivity patterns of a node in the context of its immediate neighbors. To examine the role played by each node in the context of the entire network, we calculated distributions of nodal efficiency. A node's nodal efficiency is defined as the mean inverse directed shortest path length between itself and all other nodes in the network and quantifies the ease with which that node can theoretically transmit information to all other nodes (see 'Materials and methods'). As shown in *Figure 6b*, there was a close match between the nodal efficiency distributions for the connectome and the SGPA model (with L = 0.725 mm): both exhibited an approximately normal distribution, save for a small selection of nodes with zero-valued nodal efficiency (corresponding to nodes with no outgoing connections). A degree-controlled random graph also showed approximately normally distributed nodal efficiency distributions, but the mean of the distribution was slightly higher than either the SGPA model or the mouse connectome. In contrast, the purely geometric PA model showed a sharply peaked nodal efficiency distribution, similar to what one would expect for a directed Erdos-Renyi graph (*Figure 5—figure supplement 1*, *Figure 6—figure supplement 1*). This analysis suggests that the SGPA model captures the statistical connectivity patterns of individual nodes in relation to the whole network.

Prior work has shown that a network's resilience to the removal of nodes can provide insight into its structural composition (*Newman, 2010*; *Kaiser et al., 2007*). These studies typically explore the structure of undirected networks, so for our analyses we converted the directed SGPA model to an

undirected one by ignoring directionality of edges (see 'Materials and methods'). We then simulated a lesioning (or percolation) process to compare the undirected SGPA model and other standard undirected models to the mouse connectome. When nodes (and the edges connected to them) were removed in order of decreasing degree (i.e. targeted attack), the SGPA model's global efficiency (akin to nodal efficiency; see 'Materials and methods') and largest (giant) component size both decreased in a manner more similar to the mouse connectome than any standard graph (*Figure 7*). However, *Figure 7a* shows that the mouse connectome disintegrates the fastest, and *Figure 7b* shows that the global efficiency falls more rapidly in the connectome than any model. Thus, the connectome appears more vulnerable to targeted attack than the model networks explored here.

The density of edges can influence some metrics (like global efficiency). To test if our lesioning results were a product of connection density (which is itself determined by degree distribution), rather than connectivity patterns specific to each model, we also investigated the connection density throughout the lesioning process (*Figure 7—figure supplement 1a*). We found that, throughout the lesioning process, the connectome's connection density was similar to that of all models except the small world network (and necessarily matched the degree-controlled random graph). We also found that while global efficiency and connection density were correlated, their specific relationship depended on the model (*Figure 7—figure supplement 1b*). Interestingly, the way these two variables were related in the connectome most resembled how they were related in the SGPA model, relative to all the others. This shows that the results in *Figure 7* are not simply due to differences in degree distribution and the dependence of global efficiency on connection density. Therefore, we propose that these lesioning results expose differences in higher order connectivity structure between the connectome and the examined models. Average global efficiency and other undirected metrics are shown in *Table 1*.

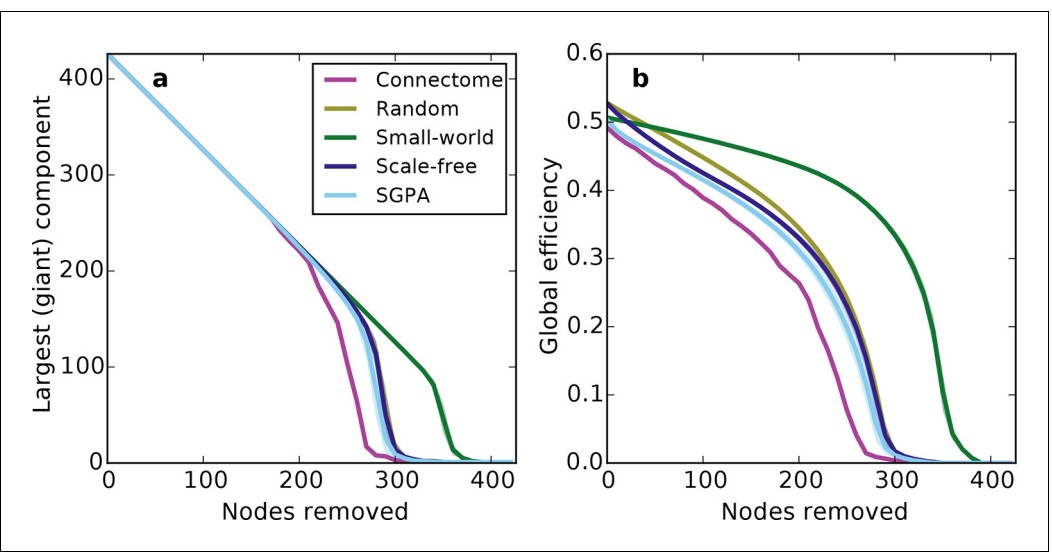

**Figure 7.** Response of undirected networks to targeted lesions where nodes are removed in order of highest degree. Mean values (lines) ± standard deviations (shaded regions) are plotted for each model after 100 repeats. (a) Size of the largest (giant) connected component in response to targeted attack. The randomly shuffled connectome (Random) is obscured by the scale-free graph. (b) Global efficiency for the mouse connectome and network models in response to targeted attack.

The following figure supplement is available for figure 7:

**Figure supplement 1.** Network connectivity patterns (and not just connection density) affect global efficiency.

**Table 1.** Average undirected metrics for the connectome, standard, and model networks. For the random, small-world, scale-free, and SGPA models, the standard deviation for 100 repeats is also shown. See Table 2 for metric definitions. Note that properties for the original (single-hemisphere) connectome are presented in **Oh et al. (2014)**.

|  | Clustering | Characteristic path length | Global efficiency |
|---|---|---|---|
| Connectome | 0.361 | 2.226 | 0.492 |
| Random | 0.140 ± 0.0013 | 2.002 ± 0.0020 | 0.528 ± 0.0003 |
| Small-world | 0.359 ± 0.0058 | 2.129 ± 0.0058 | 0.507 ± 0.0010 |
| Scale-free | 0.159 ± 0.0037 | 1.998 ± 0.0026 | 0.527 ± 0.0004 |
| SGPA | 0.343 ± 0.0094 | 2.166 ± 0.0159 | 0.501 ± 0.0027 |

## Discussion

The Allen Mouse Brain Connectivity Atlas (*Oh et al., 2014*) provides a unique view into the mesoscale structure of the mammalian brain. Through anatomical tracing experiments using genetically identical mice registered to a common reference frame, this dataset permitted a graph theoretic analysis of whole-brain connectivity. We first showed that the connectome's lower degree nodes are more clustered – a relationship absent in the standard random graphs used in the literature. We then demonstrated that in-degree is approximately normally distributed and out-degree is approximately exponentially distributed in the connectome; because in- and out-degree were uncorrelated, this meant that nodes with more total connections tended to have a higher percentage of outgoing connections. Additionally, we found that reciprocal edges were shorter than nonreciprocal edges on average and that the proportion of reciprocal edges was substantially higher in the connectome than that expected by chance. Finally, we developed a directed generative network model based on two simple rules: SG and PA. This directed model captures many (but not all) spatial and graph theoretic properties of the mouse connectome and provides important biological insights into the organizational principles governing neural connectivity at the mesoscale, which we discuss later.

### The importance of directed analysis

The fact that in- and out-degree were differentially distributed in the connectome highlights an important limitation of undirected graphs: they do not discriminate between in- and out-degree distributions and may therefore fail to reveal key connectivity properties arising from such directed structure. Interestingly, the in- and out-degree relationships observed at other scales and in other organisms differ from our findings here. For instance, the *C. elegans* connectome (for the full nervous system) (*White, 1985*) has nearly identical in- and out-degree (exponential) distributions (*Amaral et al., 2000*). While numerous studies have found common properties across scales (e.g. small-worldness), more pronounced differences in network structure, such as different in- and out-degree distributions, suggest that a single set of generative rules cannot capture the brain's network structure across different scales and/or organisms.

### Indications of functional segregation and integration in the connectome

Functional segregation and integration comprise two components of a framework commonly used to interpret brain network architecture (*Rubinov and Sporns, 2010*; *Tononi et al., 1994*). Functional segregation asserts that the brain carries out specialized computations in anatomically localized and highly interconnected regions. As high clustering coefficient is thought to indicate potential for participation in this sort of computational unit (*Rubinov and Sporns, 2010*), our finding that nodes with a small number of neighbors tend to be highly clustered suggests that these nodes are candidates for specialized processing. However, functional data would be required to validate this conjecture since high clustering coefficient does not necessarily imply grouping of nodes into specialized clusters. Functional integration predicts that some high-degree "hub" brain regions consolidate the results of these specialized computations for higher function (e.g. as in behavior and perception) (*Friston, 2002*).

Surprisingly, connections associated with such hub nodes were primarily outgoing in the directed connectome (*Figure 3b*). Assuming again that anatomical connectivity is indicative of functional relationships, our results suggest that hubs play a stronger role in distribution, rather than integration, of information. However, these ideas are not necessarily contradictory – it is possible that many brain regions are involved in both integration and distribution to varying extents. In fact, the proportion in-degree distribution roughly resembles a uniform distribution (*Figure 3—figure supplement 1*) suggesting a continuum of these properties rather than discrete categories.

Above-chance reciprocity, as we found in the mouse connectome, is also hypothesized to be important for functional integration and segregation (*Tononi et al., 1994*). Previous work has suggested that bidirectional connections between brain regions constrain the dynamics in such a way that a balance arises between the two (*Tononi et al., 1992*; *Sporns et al., 1991*). Considering the propensity for feedback within the brain, however, perhaps the relatively high reciprocity is not surprising. Like the asymmetric degree distributions, the connectome's high reciprocity also highlights the importance of using directed graph models (as opposed to undirected ones) when analyzing brain networks.

## Organizational principles of the mouse connectome

The generative rules explored here follow from a series of recent publications attempting to develop generative network models of animal connectomes (*Song et al., 2014*; *Kaiser et al., 2009*; *Ercsey-Ravasz et al., 2013*; *Klimm et al., 2014*; *Betzel et al., 2015*; *Lim and Kaiser, 2015*). Several papers have documented the importance of PA in generative network models of animal connectomes (*Song et al., 2014*; *Kaiser et al., 2009*; *Ercsey-Ravasz et al., 2013*; *Klimm et al., 2014*; *Kaiser and Hilgetag, 2004*). For example, *Kaiser et al. (2009)* noted that if axonal outgrowth occurs in a straight line and the axon attaches to the first node it physically encounters, then the probability of connecting to a target neuron depends exponentially on the target's distance from the source neuron (*Kaiser et al., 2009*). Previous research has also shown that some chemical gradients responsible for axon guidance decay exponentially with distance (*Murray, 1993*; *Isbister et al., 2003*) and have been modeled as such (*Mortimer et al., 2009*). Similarly, *Rubinov et al. (2015)*, recently reported that distance-dependent inter-areal connection strengths in the mouse connectome were best captured by a power law. We explored a purely geometric model based on PA and found that while this model induces clustering between low-degree nodes, it fails to account for the broad degree distribution of the connectome. This motivated the exploration of two mathematically symmetric generative models that included an additional topological rule: target attraction proximal attachment (TAPA) and source growth proximal attachment (SGPA).

Only when incorporating the source-growth rule could we capture the in- and out-degree distributions of the mouse connectome. Note that these distributions arise independently of the PA rule, but other characteristics require it, as discussed previously (see *Figure 5—figure supplement 2* for the in- and out-degree distributions of a graph with source-growth but not PA). To explore whether source selection based specifically on out-degree was a necessary property, we also examined a network grown with source selection probability proportional to the *total* degree raised to a power. While this model also exhibited many of the connectome's properties, it makes predictions that are not observed in the connectome (e.g. it predicts that the number of incoming and outgoing connections should be correlated; *Figure 5—figure supplement 5*); the model itself also requires an additional parameter. This indicates that a SG rule specifically depending on out-degree is a better candidate mechanism for generating the properties of mouse connectome.

Our results indicate that much of the organizational structure of the mouse connectome is captured by the geometric and topological generative rules employed by the SGPA model. The ability of these simple rules to closely capture the mouse connectome's network structure raises the possibility that brain organization at the mesoscopic scale does not require precise specification of connectivity (e.g. through genetic or transcriptional factors), but might instead be largely based on a set of relatively simple instructions. Many generative schemes proposed previously, such as the homophilic rules by *Betzel et al. (2015)* (nodes with similar graph theoretic properties are more likely to connect), or the minimal wiring length networks by *Klimm et al. (2014)*, are also based on simple connectivity rules, but these rules are not readily interpretable in terms of biophysical processes. We previously discussed possible biological underpinnings of the geometric PA rule, but the topological SG rule also lends itself to speculation on a set of possible biological mechanisms.

## A biophysical interpretation of source growth

The source growth rule might be realized during brain development by the actions of a family of proteins known as neurotrophins, which play a major role in promoting the survival of innervating neurons (*Huang and Reichardt, 2001*; *Chao, 2003*) and the growth and branching of their axons (*Tessier-Lavigne and Goodman, 1996*). Found throughout both the central and peripheral nervous systems, these proteins are typically secreted by a target (postsynaptic) neuron at functional synapses, endocytosed by a source (presynaptic) neuron, and retrogradely transported to the source neuron's soma, where they trigger the above-mentioned processes (*Huang and Reichardt, 2001*; *Korsching, 1993*). Because neurotrophins are available in limited quantities, they are hypothesized to cause competitive interactions among growing neuronal populations (*van Ooyen and Willshaw, 1999*). Indeed, such an interaction resembles the way in which a single source node is probabilistically selected for each edge addition in the SGPA model (as opposed to allowing all nodes to generate outgoing connections simultaneously). Most importantly, since populations that maintain many functional outgoing connections will tend to receive more neurotrophins, they should on average be more fit for survival and new connection generation, and their existing connections should be more likely to branch. This would in turn increase their probability of establishing new connections to novel targets. Such a biological mechanism would lead to the "rich-get-richer" phenomenon for outgoing connections that was fundamental in modeling the network properties of the mouse connectome.

## A non-power law explanation of the connectome's undirected degree distribution

A final insight provided by the SGPA model concerns the undirected degree distribution in the mouse connectome. Degree distributions in the brain have often been characterized by power laws, where $p(k) \propto k^{\gamma}$ for some $\gamma < 0$ (*Bullmore and Sporns, 2009*; *Kaiser et al., 2007*). The SGPA model provides an alternative because the in- and out-degree distributions are driven by different rules. In-degree is approximately Gaussian because of the central limit theorem (connections are random in the sense that they only depend on distance and not degree or other topological properties), and out-degree is approximately exponential, owing to the SG process. If in- and out-degree are independent, as they are in our model and seem to be in the connectome, the distribution of their sum (i.e. of total degree) will be the convolution of the two individual distributions. Thus, we would expect the distribution over undirected degree (which is approximately the sum of in- and out-degree, the approximation arising since we do not count reciprocal edges twice) to approach the convolution of a Gaussian and an exponential, that is, an exponentially modified Gaussian (*Figure 6—figure supplement 2*). Additionally, the SGPA model is able to capture the low-degree portion of the distribution (*Figure 1b*) unlike the scale-free graph. This serves as a more biologically motivated alternative to the power-law degree distribution often used to describe the brain's network properties.

## Limitations of the SGPA model

While the SGPA model captured a number of the connectome's characteristics, there are also limitations worth highlighting. Our choice to instantiate all the nodes at the start of the network's generation is somewhat biologically implausible. This is different from the Barabasi-Albert algorithm used to generate scale-free graphs where, aside from an initial group of nodes, all the nodes are added sequentially over time. However, a version of our SPGA model in which nodes *are* added one-by-one (see 'Materials and methods') yielded a qualitatively similar network (see *Figure 5—figure supplement 3* and *Figure 6—figure supplement 3*). The only notable differences were that the youngest nodes had an extremely low degree and a clustering coefficient of either zero or one, and that there was a positive correlation between in- and out-degree. This deviation from the properties observed in the mouse connectome could potentially be corrected by "pruning" the final network or by modifying the probability of generating new connections as the final nodes are added.

In our simulations, we chose to randomly draw node positions from a 7 mm$^3$ cube to spatially embed our model. The dimensions of this cube were chosen to match the inter-nodal distance distribution of the connectome (*Figure 5—figure supplement 4*). All the simulations we explored, however, yielded comparable results when run with node coordinates determined by the centroids of brain regions in the connectome. One interesting feature of the SGPA model is that it predicts that

edge lengths should be significantly shorter than those observed in the connectome. This is a consequence of fitting the length constant to the connectome's reciprocity. A more satisfactory fit to the edge length distribution can be obtained by using a larger length constant, but this substantially reduces the reciprocity and degrades the inverse relationship observed between clustering and degree (*Figure 6a*). Equivalently, we can scale the dimensions of the model, which allows us to match the reciprocity and the edge length distribution but not the inter-nodal distance distribution. Thus, our model is unable to account for both the edge length distribution and the reciprocity of the connectome while also maintaining the appropriate dimensionality.

Within the lesioning analysis, we found that the connectome is more susceptible to targeted attack than our generative SGPA model network (or any other model). This suggests that the mouse connectome is less resilient than the synthetic networks explored here. More broadly, addressing whether the SGPA model can reproduce the macroscale hierarchical modularity recently reported by *Rubinov et al. (2015)* is also a target for future research. Previous work on targeted attacks on the macaque and cat macroscale connectome has shown different patterns of results (*Kaiser et al., 2007*), but it is difficult to assess whether this is due to differences in the organism (mouse vs. monkey or cat), scale (meso vs. macro), or both.

Lastly, we treated the entire connectome as a network grown with homogeneous growth rules. The success of our approach lends some merit to this assumption, but it is nevertheless likely that cortico-subcortical connections follow different generative rules than cortico-cortical ones, for example. A similar point was raised by *Kaiser and Varier (2011)* and *Oh et al. (2014)* – both noted that cortico-cortical projections have higher reciprocity than cortico-subcortical ones in macaque macroscale and mouse mesoscale connectomes, respectively. Uncovering the differences in generative rules employed by subnetworks in the brain, as well as those at different scales or time points, is a target for future research. Dynamically changing the physical scale of the model (e.g. to simulate physical growth) is another promising avenue for future work which some have begun to explore (*Ozik et al., 2004*), and may account for our inability to capture the edge length distribution in the connectome. Here, our network was embedded within a physical space that maintained a constant size. We did find that a node's age affected its final properties (*Figure 5—figure supplement 3*, *Figure 6—figure supplement 3*), but the growth of neural tissue typically occurs in an expanding physical space which may stretch or alter the oldest connections as development proceeds. Continued use of real data to both inspire and evaluate network models will be crucial for elucidating the principles that govern the network organization of connectomes.

## Conclusions

We have characterized the network properties of the mouse mesoscale connectome, a system that highlights the importance of using spatial, directed graphs when modeling brain networks. A model that uses two simple organizational principles – source growth and proximal attachment – can capture a large number of directed, undirected and spatial network properties of the mouse connectome. Importantly, these rules have biologically plausible connections to developmental mechanisms and wiring properties in real brains. This model not only serves as a simple mathematical tool that can be used to model and understand mesoscopic brain organization, but also provides a parsimonious framework for informing future investigations of brain network formation.

## Materials and methods

All code used in this analysis is available at *https://github.com/neofunkatron/neofunkatron*

### Empirical connectivity graph

To generate a connectivity graph from empirical data, we used the mouse connectome developed by the Allen Brain Institute. To compile this dataset, anterograde tracers were injected in a single hemisphere and the authors traced projections into *both* hemispheres. Thus, the connectome probed connections originating in the right hemisphere and terminating in either the right or left hemisphere. The original work computed projection strengths and associated p-values (the probability that the observed projection would arise by chance) between 213 pairs of ipsi- and contralateral anatomically defined brain regions (*Oh et al., 2014*). To generate a binary adjacency matrix, we set all connection strengths with p<0.01 to one and all other elements to zero. This yielded a graph

with a reasonable connection density for analysis. We then assumed that all connectivity projections were bilaterally symmetric in order to construct a 426 x 426 binary adjacency matrix, which completely defines a (single component) graph. The results remained qualitatively the same for alternative connection thresholds (p<0.05 and p<0.001).

### Random graphs

The following sections describe the random graphs we included in our analysis. For each random graph, the number of nodes was set to 426, and the number of edges to 7804 (for undirected models) or 8820 (for directed models).

### Standard random graphs

Standard random graphs were generated in Python using the NetworkX module (https://networkx.github.io) (Hagberg et al., undefined). The small-world graph (*Watts and Strogatz, 1998*) is parameterized by the number of nodes *n*, the number of initial connections for each node $k_{SW}$, and the probability of rewiring, *p*. Here, we used $k_{SW}$ = 18 to match the mean degree of the connectome, and p=0.23 to approximately match the mean clustering coefficient of the connectome. The scale-free graph (*Barabasi and Albert, 1999*) is parameterized by the number of nodes *n* and the number of connections $k_{SF}$ formed by each node as it is added to the network. We used $k_{SF}$ = 18 to match the number of edges in the connectome. We generated the undirected degree-controlled random network by shuffling the mouse connectome's edges while holding the degree distribution constant (similar to *Maslov and Sneppen, 2002*); this generates a control graph with random connectivity but identical degree distribution. The directed degree-controlled random was generated in a similar way, except both the in- and out-degree distributions were held nearly fixed ("nearly" because the algorithm often converts a small number of edges to self- or double-connections, which are ignored in our analysis; however, these only represent about 5% of all connections). These standard graphs are similar to those used in *Oh et al. (2014)*. The directed Erdos-Renyi graph was generated using NetworkX, with an edge probability of 0.0487, which on average yielded the number of directed edges present in the mouse connectome.

### Custom random graphs

The following sections describe the algorithms used to generate the custom random graphs examined in this study.

### Purely geometric models

For the purely geometric PA graph, 426 nodes were randomly assigned centroids within a 7 mm x 7 mm x 7 mm cube, with no edges connecting the nodes. We then generated each *undirected* edge by first selecting a source node *i* at random and subsequently selecting a target node *j* with probability $P_{ij} \propto exp(-d_{ij}/L)$, where $d_{ij}$ denotes the distance between node *i* and node *j*. If an undirected edge already existed between the source and target, the probability of selecting that target was set to zero. Edges were added until the number of (undirected) edges matched that of the connectome (K=7804). The directed PA algorithm was identical, except that directionality of edges was retained. For the graphs incorporating topological rules (i.e. SGPA and TAPA), the growth algorithm was initialized by instantiating *n* nodes with no edges connecting them, except for one self-connection per node to prevent zero-valued connection probabilities for nodes with no outgoing or incoming edges (for SGPA and TAPA, respectively). However, these self-connections were ignored when calculating all metrics for the final graph. In all the graphs we generated (except for node-by-node SGPA), we matched *n* and *K* to the number of nodes and directed edges, respectively, in the empirical connectome (*n* = 426, *K* = 8820).

### TAPA random graph

At each step in the growth process an edge is added by (1) selecting the target node *j* from all nodes without maximum in-degree according to the in-degree of *j*: , and (2) selecting the source node *i* with a probability that decreases with distance from the target: $P(source = i|target = j) \propto exp(-d_{ij}/L)$. If a connection already existed from node *i* to *j*, this probability was set to zero to avoid a duplicate edge. $d_{ij}$ is the Euclidean distance between nodes *i* and *j*,

and *L* is a parameter governing the strength of distance-dependence. For the nonspatial TA graph (*Figure 5—figure supplement 2*), the algorithm was as the TAPA algorithm, except the source was chosen with a uniform probability for all nodes (i.e. no distance-dependence).

### SGPA random graph

At each step in the growth process an edge is added by (1) selecting the source node $i$ according to its out-degree as $P(source = i) \propto k_i^{out}$ from all nodes that do not have maximal out-degree, and (2) selecting the target node $j$ with a probability that decreases with distance from the source: $P(target = j|source = i) \propto exp(-d_{ij}/L)$. If a connection already existed from node $i$ to $j$, this probability was set to zero to avoid a duplicate edge. For the nonspatial SG graph (*Figure 5—figure supplement 2*), the algorithm was as the SGPA algorithm, except the target was chosen with a uniform probability for all nodes (i.e. no distance-dependence).

Both the target attraction and source growth rules can be conceptualized as directed spatial variants of the preferential attachment rule introduced by *Barabasi and Albert (1999)*.

### Total-degree SGPA random graphs

This graph was identical to the SGPA graph mentioned above, except that probability of selecting the source node $i$ was proportional to its total degree (in-degree + out-degree) raised to a power $\gamma$: $P(source = i) \propto (k_i^{out} + k_i^{in})^{\gamma}$. Target selection was as in the SGPA model.

### Node-by-node SGPA random graph

In this algorithm, nodes are added one at a time with positions sampled uniformly from within a 7 mm x 7 mm x 7 mm cube until the graph contains 426 nodes. Upon each node addition (starting

**Table 2.** Graph theoretical metrics used for analysis.

| Metric | Brief interpretation | Definition |
|---|---|---|
| Degree | Number of edges connected to a node. This generalizes to in- or out-degree in directed graphs, describing the number of incoming or outgoing connections for a node, respectively. | $k_i = \sum_{j \in n} a_{ij}$ <br> $N =$ set of all nodes <br> $a_{ij} = \begin{cases} 1, & \text{if edge from node } i \text{ to } j \text{ exists} \\ 0, & \text{otherwise} \end{cases}$ <br> <u>Directed versions:</u> <br> $k_i^{in} = \sum_{j \in N} a_{ji} \qquad k_i^{out} = \sum_{j \in N} a_{ij}$ <br> $a_{ij} = \begin{cases} 1, & \text{if directed edge from node } i \text{ to } j \text{ exists} \\ 0, & \text{otherwise} \end{cases}$ |
| Clustering coefficient (*Watts and Strogatz, 1998*) | Level of connectivity among nearest neighbors of node $i$ | $\dfrac{2t_i}{k_i(k_i - 1)}$ <br> $t_i =$ number of triangles that include node $i$ |
| Characteristic path length (*Watts and Strogatz, 1998*) | Mean shortest undirected path length over all pairs of nodes | $\dfrac{1}{n(n-1)} \sum_{i,j \in N, j \neq i} d_{ij}$ <br> $n =$ number of nodes <br> $d_{ij} =$ shortest undirected path from $i$ to $j$ |
| Global efficiency (*Latora and Marchiori, 2001*) | Mean inverse shortest *undirected* path length over all pairs of nodes | $\dfrac{1}{n(n-1)} \sum_{i,j \in N, j \neq i} \dfrac{1}{d_{ij}}$ <br> $d_{ij} =$ shortest undirected path from $i$ to $j$ |
| Nodal efficiency (generalized from [*Achard and Bullmore, 2007*]) | Mean inverse shortest *directed* path length from a single node to all other nodes | $\dfrac{1}{(n-1)} \sum_{j \in N, j \neq i} \dfrac{1}{d_{ij}}$ <br> $d_{ij} =$ shortest directed path from $i$ to $j$ |
| Reciprocity coefficient | Proportion of edges from node $i$ to node $j$ that have a reciprocal connection from node $j$ to node $i$ (when $i \neq j$) | $\dfrac{1}{N_e} \sum_{i,j \in N, j \neq i} a_{ij} a_{ji}$ <br> $a_{ij} = \begin{cases} 1, & \text{if directed edge from } i \text{ to } j \text{ exists} \\ 0, & \text{otherwise} \end{cases}$ <br> $N_e =$ total number of undirected edges |

with the second node addition), each existing edge "branches" with a probability p=0.016 (which was chosen so as to yield approximately the same number of edges as in the connectome). When an edge "branches" a new edge is formed that has the same source node and whose target node is chosen with probability $P(target = j|source = i) \propto exp(-d_{ij}/L)$ (i.e. in the same way as in the standard SGPA model). If an edge already exists between the source node and the selected target node, no new edge is added. When all the edges and nodes have been added, any nodes that are not connected to the largest (giant) graph component (typically only three or four, representing < 1% of the network) are removed from the graph.

## Undirected metrics and lesioning of directed graphs

See *Table 2* for metric definitions. We calculated undirected metrics (e.g. clustering coefficients and undirected degree) and carried out a lesioning study for directed graphs by first casting the directed graph to an undirected one. This yielded a graph which contained an undirected edge between every pair of nodes that had been connected by at least one directed edge in the directed graph.

## Acknowledgements

We thank the Allen Institute for Brain Science for generously providing the dataset used in this analysis and for funding the Summer Workshop on the Dynamic Brain – the course in which this project was conceived. We also thank the directors of this course, Christof Koch and Adrienne Fairhall. Additionally, we are grateful to Marcus Kaiser, Martha Bosma, and the reviewers for helpful comments and suggestions.

## Additional information

### Funding

The authors did not receive financial support for this work.

### Author contributions

SH, RP, MW, Conception and design, Acquisition of data, Analysis and interpretation of data, Drafting or revising the article, Contributed unpublished essential data or reagents

### Author ORCIDs

Sid Henriksen, http://orcid.org/0000-0002-4335-4218

Rich Pang, http://orcid.org/0000-0002-2644-6110

Mark Wronkiewicz, http://orcid.org/0000-0002-6521-3256

## Additional files

### Major datasets

The following previously published dataset was used:

| Author(s) | Year | Dataset title | Dataset URL | Database, license, and accessibility information |
|---|---|---|---|---|
| Oh SW, Harris JA, Ng L, Winslow B, Cain N, Mihalas S, Wang Q, Lau C, Kuan L, Henry AM, Mortrud MT, Ouellette B, Nguyen TN, Sorensen SA, Slaughterbeck CR, Wakeman W, Li Y, Feng D, Ho A, Nicholas E, Hirokawa KE, Joines BP, Peng H, Hawrylycz MJ, Phillips JW, Hohmann JG, Wohnoutka P, Gerfen CR, Koch C, Bernard A, Dang C, Jones AR, Zeng H | 2014 | Data from: A mesoscale connectome of the mouse brain. Supplementary Data 3 | http://www.nature.com/nature/journal/v508/n7495/extref/nature13186-s4.xlsx | Available from Nature |

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
