## [Decision Letter]

Thank you for submitting your work entitled "A simple generative model of the mouse mesoscale connectome" for consideration by *eLife*. Your article has been reviewed by two peer reviewers, and the evaluation has been overseen by Heidi Johansen-Berg as the Reviewing Editor and David Van Essen as the Senior Editor.

The reviewers have discussed the reviews with one another and the Reviewing Editor has drafted this decision to help you prepare a revised submission.

Summary:

In this submission the authors tackle the important topic of generative models for neural systems. Novel features, compared to previous models for mouse and human connectome, are: (a) mechanisms for connection establishment that do not rely on any topological information about the target node, (b) mechanisms that can reproduce the distribution of uni-directional vs. bi-directional connections (reciprocity), and (c) a better link to the underlying biological mechanisms that could implement the observed connection establishment rules.

We recognize that the manuscript provides many significant novel advances but we highlight two key issues that need to be addressed in a revision.

Essential revisions:

1) As the authors note, a number of recent studies have also made contributions to this area. The take-home message from these studies has been that including geometric (spatial) information about a network's embedding in a generative model will tend to result in better-fitting synthetic networks. This highlights a weak point of the submission. The authors initially propose two growth mechanisms ("source growth" and "target attraction"), both of which form links according to topological relationships only rather than geometric or geometric + topological relationships. When the authors add geometric information to the model they observe that the synthetic networks generated by the model more closely resemble the mouse connectome. This is hardly a surprising result.

As an improvement, we recommend that the authors test at least one purely geometric generative model. This is an essential step, because there is an expectation that geometry plays an important role in shaping connectomes and the authors should assess whether geometry, alone, can generate realistic-looking synthetic networks. Next, the authors should test more complicated wiring rules, similar to the one “SGPA” model that combine geometry + topology to generate links. Any contribution above and beyond that of the purely geometric model could be attributed to the topological component of the wiring rule. This is an approach similar to the references (Betzel et al., Neuroimage, 2015, Vertes et al., PNAS, 2012).

2) There is one difference between the mouse connectome and the generated networks that might explain some of the observed differences. From the methods, it seems that there are no connections between the two hemispheres in the mouse connectome, that means there are two components with 213 nodes each (subheading “Empirical connectivity graph”). For the generated networks, on the other hand, the there is a giant component that contains almost all of the 426 network nodes. This difference will influence the calculation of the characteristic path length and of the global efficiency and might therefore influence the robustness results presented in Figure 7. Moreover, they will also affect the higher modularity in the mouse connectome as the relative proportion of edges that exists within each hemisphere (component) will be higher than the edge density with the giant component of the generated networks. There are several ways how this could be fixed: one might assume connections between hemispheres within the mouse connectome but this is difficult if there is no experimental data to back this up. Or one might use generated networks that consists of 213 nodes and then duplicate that network (like the mirroring for the mouse) to generate a network with 426 nodes and two dis-connected components. In that way, both types of networks would have the same starting configuration.

---

## [Author Response]

*In this submission the authors tackle the important topic of generative models for neural systems. Novel features, compared to previous models for mouse and human connectome, are: (a) mechanisms for connection establishment that do not rely on any topological information about the target node, (b) mechanisms that can reproduce the distribution of uni-directional vs. bi-directional connections (reciprocity), and (c) a better link to the underlying biological mechanisms that could implement the observed connection establishment rules.*

*We recognize that the manuscript provides many significant novel advances but we highlight two key issues that need to be addressed in a revision.*

We thank the reviewers for their comments. The major changes in this revision concern the reviewers’ suggestion to incorporate a purely geometric model as a useful reference for the topological models. To best accommodate this addition, we have also restructured the manuscript slightly and have replaced the non-geometric topological models (i.e. source growth and target attraction without “proximal attachment”) with geometric versions. We have also clarified our description concerning how the connectome was generated, updated the lesioning analysis, and revised the discussion on functional segregation and integration.

*Essential revisions:*

*1) As the authors note, a number of recent studies have also made contributions to this area. The take-home message from these studies has been that including geometric (spatial) information about a network's embedding in a generative model will tend to result in better-fitting synthetic networks. This highlights a weak point of the submission. The authors initially propose two growth mechanisms ("source growth" and "target attraction"), both of which form links according to topological relationships only rather than geometric or geometric + topological relationships. When the authors add geometric information to the model they observe that the synthetic networks generated by the model more closely resemble the mouse connectome. This is hardly a surprising result. As an improvement, we recommend that the authors test at least one purely geometric generative model. This is an essential step, because there is an expectation that geometry plays an important role in shaping connectomes and the authors should assess whether geometry, alone, can generate realistic-looking synthetic networks. Next, the authors should test more complicated wiring rules, similar to the one “source growth-proximal attachment” model that combine geometry + topology to generate links. Any contribution above and beyond that of the purely geometric model could be attributed to the topological component of the wiring rule. This is an approach similar to the references (Betzel et al., Neuroimage, 2015, Vertes et al., PNAS, 2012).*

We agree that a network generated with purely geometric rules would improve the manuscript, so we have made changes accordingly. Figure 2 now shows the degree distribution, as well as the clustering-degree distribution for a purely geometric model with three different values of the length parameter. We have elected to keep this network undirected as the in- and out-degree distributions will necessarily look the same for a purely geometric model (both approximately Gaussian). A key insight here is that the degree distribution of the purely geometric model is too narrow to capture the joint clustering-degree distribution of the connectome. This limitation indicates that geometric rules alone cannot capture the connectome’s connectivity properties and provides a natural transition to the next part of the manuscript where we explore the directed structure of the connectome and geometric-topological models.

We share the reviewers’ view that the manuscript would be strengthened by more alternative models, so we evaluated additional network models. First, we added a mathematically symmetric version of the SGPA model (TAPA), which incorporates both geometric and topological properties. The in- and out-degree distributions remained inverted from the true distribution although the connectome’s reciprocity was more closely emulated. Second, we explored a model in which outgoing connection probability depends on total degree, rather than out-degree. With this model, some qualitative properties of the brain were recoverable only once connection probability was raised to the powerγ. However, this model had a reliable correlation between in- and out-degree, which we do not observe in the connectome or in the SGPA model. We feel this provides additional evidence that the SGPA model captures nuanced features of the generative algorithm of the mouse connectome ‒ namely, that the growth process depends specifically on out-degree. Finally, we would like to point out that the original (and revised) manuscript did indeed explore a generalization of SGPA in which nodes were added one-by-one to the graph, rather than initialized all at once, and this graph showed similar properties to both SGPA and the connectome.

Concerning the suggested reference models, the success of the SGPA model and Figure 3 strongly suggest that in the brain, mesoscale connections are generated primarily as a function of source properties (as well as the distance between the source and target). Any model attempting to emulate the connectome must have generative rules that operate differently on incoming and outgoing connections. Therefore, we believe that it is only worthwhile to test more complicated models that can, at least in principle, incorporate this property. This rules out all undirected models, such as those proposed by Betzel et al. (2015) and Vertes et al. (2012), as well as directed models in which incoming and outgoing connection probabilities are not treated differently. Directed generalizations of other popular undirected graphs hypothesized to account for brain connectivity might prove useful in this domain, but we believe that the in-depth exploration needed to properly evaluate them is beyond the scope of this work.

*2) There is one difference between the mouse connectome and the generated networks that might explain some of the observed differences. From the methods, it seems that there are no connections between the two hemispheres in the mouse connectome, that means there are two components with 213 nodes each (subheading “Empirical connectivity graph”). For the generated networks, on the other hand, the there is a giant component that contains almost all of the 426 network nodes. This difference will influence the calculation of the characteristic path length and of the global efficiency and might therefore influence the robustness results presented in Figure 7. Moreover, they will also affect the higher modularity in the mouse connectome as the relative proportion of edges that exists within each hemisphere (component) will be higher than the edge density with the giant component of the generated networks. There are several ways how this could be fixed: one might assume connections between hemispheres within the mouse connectome but this is difficult if there is no experimental data to back this up. Or one might use generated networks that consists of 213 nodes and then duplicate that network (like the mirroring for the mouse) to generate a network with 426 nodes and two dis-connected components. In that way, both types of networks would have the same starting configuration.*

We apologize for the confusion here. During the experiments used to generate the mouse connectome, the viral tracer was indeed only injected into brain regions within a single hemisphere. The two-photon tomography system, however, imaged both hemispheres when recording the projections extending from the injected regions. That is, the mouse connectome dataset does contain interhemispheric connections. When mirroring the connectome across the sagittal midline, this results in a network comprised of a single component connecting all 426 nodes. To remedy this confusion, we have added additional explanation to the Methods section concerning the experimental procedures used in Oh et al., 2014.